# Effect of Ball Milling on the Absorption Properties of Fe_3_O_4_

**DOI:** 10.3390/ma13040883

**Published:** 2020-02-17

**Authors:** Yi Liang, Yue Yuan, Yuwei Huang, Yujiang Wang, Shicheng Wei, Bo Wang, Wei Huang, Wei Xin, Xinlei Wang

**Affiliations:** National Key Laboratory for Remanufacturing, Army Academy of Armored Forces, Beijing 100072, China; liangyi365@126.com (Y.L.); yyyue11111@163.com (Y.Y.); hywhuj1607@126.com (Y.H.); HW949963@163.com (W.H.); xw1390332928@126.com (W.X.); wxl111210026@163.com (X.W.)

**Keywords:** ball milling time, microstructure, morphology, electromagnetic parameters

## Abstract

FeCl_3_∙6H_2_O was used as raw material to produce Fe_3_O_4_, using the solvothermal method with ethylene glycol as the solvent. Fe_3_O_4_, with different particle sizes, was obtained via mechanical ball-milling by controlling the milling time. Effect of the milling time on the structure, morphology, and electromagnetic parameters of Fe_3_O_4_ were studied, and the absorption properties and mechanism of Fe_3_O_4_, for different milling times were analyzed. The results showed that the integrity of the original small spherical structure decreased as the ball milling time increased. Fe_3_O_4_ showed excellent microwave absorptions as the milling time reached 2 h, the reflection loss reached the maximum of −21.19 dB at 4.64 GHz as the thickness was 6.55 mm.

## 1. Introduction

Thanks to progress in science and technology, electromagnetic waves are now widely used in people’s lives. However, there are more only benefits but also problems like pollution. To reduce the impact of electromagnetic waves on the environment, research and development of absorption materials has become a popular research topic [1,2,3,4,5].

As one of the ferrites with an anti-spinel structure, Fe_3_O_4_ is a common magnetic material, which has the characteristics of low cost, simple production, and good magnetic properties [6,7,8]. Since it can generate large magnetic losses in alternating electromagnetic fields, Fe_3_O_4_ is one of the most widely used conventional absorption materials. However, due to the Snoek limitation in the high-frequency region, easy oxidation, high density, and narrow absorption frequency, the comprehensive electromagnetic wave attenuation properties of Fe_3_O_4_ are also limited. An effective way to improve electromagnetic wave absorption of Fe_3_O_4_, is to prepare materials with hollow, nanometer-sized structures [9,10,11,12]. Hollow structures can increase the attenuation of electromagnetic waves via multiple reflections of the incident electromagnetic waves within the cavity. Due to the high proportion of atoms on the surface of the particles, nanomaterials are prone to interface polarization, which can cause multiple scattering. Moreover, nanomaterials show quantum size effects, which split the electron energy levels of particles, and the splitting interval corresponds to the energy range of electromagnetic waves, which opens a new absorption channel [13,14,15].

Mechanical ball milling is the simplest and most used method to prepare micro/nano particles. It refers to the method of placing a material into a ball mill and grinding the material to produce broken particles and fine particles via reciprocal action between material and grinding balls. This method is characterized by a simple process and high yield, which is usually divided into two types: dry milling and wet milling. Wet milling can easily grind the product to a fine size and produce more uniform nanoscale particles.

At present, chemical synthesis is the most common method to prepare electromagnetic wave absorbers with different micro/nano sizes. However, the preparation method is difficult, the required time is long and the yield is low. In addition, the size of the prepared materials is uncertain. Therefore, it is expected to obtain particles with different particle diameters using the simplest mechanical ball-milling method, which is based on the chemical synthesis method. In this paper, the Fe_3_O_4_, absorption material was synthesized using the hydrothermal method, and the ball milling method was used to study the changes of phase, morphology, structure, and electromagnetic parameters of Fe_3_O_4_, at different ball-milling times. The effects of structure and size of Fe_3_O_4_ absorption material on the electromagnetic-wave attenuation performance were analyzed, and the feasibility of preparing nano-absorption-materials using mechanical ball-milling was discussed.

## 2. Experimental

### 2.1. Preparation of Fe_3_O_4_

(1) Preparation of the Fe_3_O_4_ absorption material: 45 ml of ethylene glycol was placed in a beaker, then, 4 g of urea and 2 g of polyvinyl pyrrolidine were added, and evenly dispersed ultrasonically. Subsequently, 1–2 g of FeCl_3_∙6H_2_O was weighed, using an electronic balance, and added to the above solution. Then, ultrasonic treatment was preformed until the FeCl_3_∙6H_2_O was uniformly dispersed in the solution. Next, the solution was placed in a 100 ml high-pressure autoclave containing polytetrafluoroethylene, and the reaction was performed at 200 °C for 12 h. After the reaction, the sample was taken out, washed with anhydrous ethanol and distilled water many times, and dried for 24 h in a vacuum drying oven.

(2) Ball milling treatment of the prepared Fe_3_O_4_: To prevent the introduction of impurities caused by the abrasion of the grinding balls in the ball-mill tank during dry milling, wet milling was used. Fe_3_O_4_ samples were placed into the ball-mill tank, following the ratio of grinding ball:material:alcohol = 7:4:3, and ball milling for 0, 0.5, 1, 1.5, and 2 h, respectively. After filtration, they were placed in a vacuum-drying oven and dried for 6 h to obtain ball-milled Fe_3_O_4_ particles. The morphology, structure, surface elements, and the electromagnetic parameters were analyzed using field emission scanning electron microscope (FESEM, JEOLJSM-6500F, Eindhoven, Holland), transmission electron microscopy (TEM, Tecnai-TF20, Oberkochen, German), X-ray diffraction (XRD, D/MAX-2500PC, Rigaku, Tokyo, Japan), X-ray photoelectron spectroscopy (XPS, Thermo Fisher Scientific, Massachusetts, U.S.A) and vector network analyzer (VNA, N5242A, Agilent, USA)SEM, TEM, XRD, XPS, and VNA, respectively. The electromagnetic parameters of the measured samples were prepared by mixing the products (60%) with molten paraffin wax (40%), and placing them into a toroidal mold (*Φ_in_* = 3 mm, *Φ_out_* = 7 mm) with a thickness of 2.5–3.0mm.

### 2.2. Testing and Characterization

A high-power turning-target polycrystalline SmartLab XRD (D/MAX-2500PC, Rigaku, Tokyo, Japan) was used, and the test condition was set as Cu target, with a scanning rate of 2°/min and a scanning range of 5–90°. The surface morphology of the samples was analyzed using FESEM (SU-8010, Hitachi, Tokyo, Japan)HitachiThe microstructure of the samples was analyzed using TEM (JEM 2100, Tokyo, Japan). The electromagnetic parameters of the samples were measured using VNA (N5242A, Agilent, Santa Clara, CA, USA), and the filling amount of the samples in paraffin was 40%.

## 3. Results and Discussion

### 3.1. Phase Analysis of Fe_3_O_4_

In order to analyze the effect of ball-milling time on the structure of Fe_3_O_4_, a polycrystalline target-turning X-ray diffraction analysis was performed. Figure 1 shows the XRD spectra of Fe_3_O_4_ at different ball-milling times. It was found that the diffraction peaks were sharp and strong, indicating that the prepared nanoparticles had high crystallinity. When the ball-milling time was 0 h, peaks at 30.2°, 35.6°, 43.2°, 53.6°, 57.1°, and 62.4° in the figure corresponded to the crystal planes of (220), (311), (400), (422), (511), and (440), respectively. According to the crystal plane, corresponding to the diffraction peak position, it can be known that the grain was Fe_3_O_4_. As the ball-milling time increased, the intensity of the diffraction peaks of Fe_3_O_4_ decreased, while the impurity peaks in the XRD spectra increased. This indicates that the milling energy increased along with the milling time, and the Fe_3_O_4_ particles broke up. Therefore, the grain size decreased and the material was further refined. However, it was also more prone to undergo oxidation. In this case, the product was a mixture of Fe_3_O_4_ and Fe_2_O_3_, and the extension of ball-grinding time affected the crystal structure of Fe_3_O_4_.

### 3.2. Effect of Ball-Milling Time on the Morphology of Fe_3_O_4_

Figure 2 shows the microstructure of Fe_3_O_4_ after different ball-milling times. It can be seen from Figure 2a, the Fe_3_O_4_, prepared using the hydrothermal method, shows a complete and regular spherical shape, with a spherical diameter 300–400 nm. With the increase in milling time, it was found, from Figure 2b–e, that the morphology of the Fe_3_O_4_ particles changed significantly. The original complete Fe_3_O_4_ pellets were constantly destroyed, and the particle size significantly reduced. After grinding for 1 h, the dispersion of Fe_3_O_4_ became worse, and agglomeration was observed. When the ball-milling time reached 2 h, it was observed that most of the Fe_3_O_4_ was ground into broken particles, and only a few intact pellets remained. The size of the broken Fe_3_O_4_ particles was 40–80 nm.

Figure 3 are the TEM images of Fe_3_O_4_ samples at different ball-milling times. From Figure 3a, it was found that the Fe_3_O_4_, prepared using the hydrothermal method, showed a complete spherical shape, with a deep edge contrast and shallow center contrast, and the Fe_3_O_4_ pellets showed a clearly hollow shape. As can be seen from Figure 3b–d, as the ball-milling time increased, the small Fe_3_O_4_ pellets were continuously broken, the integrity of the spherical particles continued to decline, and the particles became gradually fragmentated. Local agglomeration also occurred, while the size of the broken particles decreased. In general, the hollow structure of the Fe_3_O_4_ absorption material was continuously destroyed with grinding time, but the particle size of the material was continuously reduced. This is consistent with the results observed using SEM.

### 3.3. Electromagnetic Parameter Analysis of Fe_3_O_4_ at Different Ball-Milling Times

Figure 4 shows the curves of the complex permittivity and complex permeability of the Fe_3_O_4_ absorption material for different grinding times, changing with frequency. Figure 4a shows the curve of the real part of the complex permittivity of the absorption material as a function of frequency. As can be seen from the figure, the real part of the complex permittivity decreased before it increased with increasing ball-grinding time. When the ball-milling time was 1.5 h, the real part was the smallest, and when the milling time reached 2 h, the real part started to increase. Figure 4b displays the curve of the imaginary part of the complex permittivity of the Fe_3_O_4_ absorption material, as a function of frequency. It was found that the imaginary part of the complex permittivity decreased first with increasing ball-milling time and then remained basically unchanged. The value that varied with frequency was basically the same. Moreover, wave peaks appeared around 4 GHz, 9 GHz, and 15 GHz, indicating that the products had a strong dielectric loss capability at these three frequencies. The imaginary part of the complex permittivity of the Fe_3_O_4_ absorption material decreased as the ball milling time increased. This is because the originally spherical Fe_3_O_4_ particles were destroyed after ball milling. For the same mass of powder, the hollow Fe_3_O_4_ pellets (with low density) added more materials than the high dense Fe_3_O_4_ fragments, and its distribution in paraffin is also greater. Due to the increased contact area, a large number of hollow Fe_3_O_4_ spheres formed a macroscopic conductive chain or local conductive network in the material, under the action of the electromagnetic field. Therefore, in the absence of ball grinding, the absorption material with a large amount of Fe_3_O_4_ hollow pellets, had both a higher electrical conductivity and dielectric constant. As the ball-milling time increased, the small spherical structures of Fe_3_O_4_ were destroyed, which resulted in a decrease in electrical conductivity and permittivity of the material. The Fe_3_O_4_ pellets were ground when the ball-milling time reached 2 h. However, due to sufficient ball-grinding time, the size of the broken particles was smaller and it was easier to form a conductive network. The conductivity of the material showed a rising trend, and the permittivity increased significantly, especially at high frequencies.

Figure 4c,d shows the curves of the real and imaginary parts of the complex permeability of the Fe_3_O_4_ absorption material, as a function of frequency. As shown in the figures, with increasing frequency, both real and imaginary parts of the complex magnetic permeability of Fe_3_O_4_, at different ball-milling times, decreased continuously and remained unchanged afterwards. It gradually decreased with the increase of ball-milling time. This is because, when the hollow balls were ground, the material lost the advantage of the hollow structure and reduced the reflection loss of incident electromagnetic waves in the cavity. This, thus, reduced the magnetic loss capacity of the absorption material. However, the imaginary part of the magnetic permeability changed slightly with the ball-milling time, which indicates that the destruction of the hollow small sphere structure of Fe_3_O_4_ had no significant effect on the magnetic loss of the material.

The electromagnetic wave loss factor is usually used to characterize the absorption attenuation capacity of a material, and it can be described as [16]
(1)$$\tan\delta ={\tan\delta }_{E}+{\tan\delta }_{M}$$


In the formula: ${\tan\delta }_{E}$ is the tangent of electrical loss, ${\tan\delta }_{E}=\varepsilon^{\prime \prime }/\varepsilon^{\prime }$; ${\tan\delta }_{M}$ is the tangent of magnetic loss, ${\tan\delta }_{M}=\mu^{\prime \prime }/\mu^{\prime }$. Among them, ε″ and ε′ are the imaginary and real parts of the complex permittivity, and μ″ and μ′ are the imaginary and real parts of the complex permeability, respectively. It can be seen that materials with better electromagnetic-wave attenuation can be obtained by increasing the imaginary part and lowering the real part of the absorption material.

Figure 5 shows the dielectric loss tangent and magnetic loss tangent of Fe_3_O_4_ absorption material at different ball milling times. From Figure 5a,b, it can be seen that, with increasing ball-milling time, the electrical loss tangent of Fe_3_O_4_ absorption material gradually decreased in the low frequency band. However, in the high frequency band, it first decreased before it increased. This is explained as follows: During the ball milling process, the fragmentation degree of small spherical Fe_3_O_4_ absorption material improved continuously. This reduced the micro-interface of the absorption material on the whole, weakened the multiple reflection of the incident electromagnetic wave inside the material structure, and further degraded the interface polarization and dielectric loss of the Fe_3_O_4_ absorption material. On the other hand, as the milling time reached 2 h, even though the small spherical Fe_3_O_4_ was basically broken, due to the sufficient ball-milling time, the broken particles were ground into finer and more uniform nano-sized particles. Compared with other short-time grinding, the conductivity and dielectric loss of the Fe_3_O_4_ absorption materials were improved, especially in the high-frequency band.

The effect of different ball-milling times on the magnetic loss tangent of Fe_3_O_4_ absorption material was not obvious, which indicates that with the extension of the ball-milling time, although the Fe_3_O_4_ absorption material was gradually broken from the original small spherical shape into fine particles and the structure of the material changed significantly, it had no effect on its magnetic loss. Therefore, changing the microstructure of Fe_3_O_4_ absorption material via ball milling mainly affected the dielectric properties of the material.

### 3.4. Electromagnetic-Wave Attenuation Mechanisms of Fe_3_O_4_ at Different Ball-Milling Times

#### 3.4.1. Effect of Ball-Milling Time on the Absorption Mechanism

Through the analysis of electromagnetic parameters and loss factor, it was implied that the microstructure of the Fe_3_O_4_ absorption material was changed by ball milling, and its dielectric loss was greatly affected by the refining material size. To learn more about the dielectric loss, Cole–Cole diagrams were used to study the dielectric properties of Fe_3_O_4_ absorption material at different ball-milling times.

The formula for the permittivity with different frequencies [17] was proposed by K. S. Cole and R. H. Cole
(2)$$\varepsilon -\varepsilon_{\infty }=\frac{\varepsilon_{S}-\varepsilon_{\infty }}{1+{\left(j\omega \tau_{0}\right)}^{1-\alpha }}$$


Here, $\tau_{0}$, α, $\varepsilon_{\infty }$, and $\varepsilon_{S}$ represent the relaxation time, parameter variable, optical frequency permittivity and static dielectric constant, respectively. The complex permittivity ε can be expressed by as [18]
(3)$$\varepsilon =\varepsilon^{\prime }-j\varepsilon^{\prime \prime }$$


The real part $\varepsilon^{\prime }$ and imaginary part $\varepsilon^{\prime \prime }$ of the permittivity are
(4)$$\varepsilon^{\prime }=\varepsilon_{\infty }+\frac{\varepsilon_{S}-\varepsilon_{\infty }}{1+{\left(\omega \tau \right)}^{2}}$$
(5)$$\varepsilon^{\prime \prime }=\frac{\sigma_{r}+\sigma_{R}}{\omega \varepsilon_{0}}+\frac{\left(\varepsilon_{S}-\varepsilon_{\infty }\right)\omega \tau }{1+{\left(\omega \tau \right)}^{2}}$$


The Cole–Cole circular equation for the real and imaginary parts of the complex permittivity can be obtained by combining Equations (3) and (4).
(6)$${\left(\varepsilon^{\prime }-\frac{\varepsilon_{S}+\varepsilon_{\infty }}{2}\right)}^{2}+{\left(\varepsilon^{\prime \prime }-\frac{\sigma_{r}+\sigma_{R}}{\omega \tau }\right)}^{2}={\left(\frac{\varepsilon_{S}-\varepsilon_{\infty }}{2}\right)}^{2}$$


It can be seen that the center coordinates were $\left(\frac{\varepsilon_{S}+\varepsilon_{\infty }}{2},\frac{\sigma \varepsilon_{r}+\sigma_{R}}{\omega \tau }\right)$, and the radius was $\frac{\varepsilon_{S}-\varepsilon_{\infty }}{2}$. Thus, Cole–Cole diagrams of the Fe_3_O_4_ absorption material, at different ball milling times, can be obtained—see Figure 6.

According to the center coordinates and radii in Figure 6, the optical frequency permittivity $\varepsilon_{\infty }$, the static permittivity $\varepsilon_{s}$, and the conductivity σ = σr + σR can be calculated. It can be seen that the semi-circle radius and vertical coordinate of the material, after ball grinding, decreased before it increased. This indicates that the optical frequency permittivity $\varepsilon_{\infty }$ and static permittivity $\varepsilon_{s}$ decreased, after ball milling, and the electric conductivity first decreased and then increased, which is consistent with the results of the dielectric-loss analysis. Therefore, although the dielectric loss of the Fe_3_O_4_ absorption material can be improved using a sufficiently long ball-milling time, the overall dielectric-loss decreases more than that for the hollow small spherical Fe_3_O_4_ without ball grinding. Its absorption mechanism was the same as the polarization relaxation before ball milling, and both involve interfacial polarization and dipole polarization relaxation loss.

The loss mechanisms of magnetic loss materials mainly include hysteresis loss, eddy current loss, domain-wall resonance, natural resonance, exchange resonance, and others. In general, hysteresis loss is small in weak external magnetic fields, and domain-wall resonance occurs for the range of 1–100 MHz. Using Aharroni’s theory, when one dimension of a nanomaterial is reduced to the nanometer level, it may generate a resonance mode with higher resonance than the natural resonance. The exchange resonance has been confirmed in many studies. The particle size of the Fe_3_O_4_ absorption material, prepared via ball milling, was within the nanometer range. Hence, there was some exchange resonance loss. To study the loss of magnetic absorption materials, the following formula is generally used [19]
(7)$$\mu^{\prime \prime }{\left(\mu^{\prime }\right)}^{-2}f^{-1}=\frac{2}{3}\pi \mu_{0}d^{2}\sigma$$


Here, σ is the conductivity of the material, and $\mu_{0}$ is the vacuum permeability. In other words, if there is only eddy-current loss, the right side of the formula should be constant. Based on the calculation of electromagnetic parameters, the $\mu^{\prime \prime }{\left(\mu^{\prime }\right)}^{-2}f^{-1}$ vs. frequency curve of the Fe_3_O_4_ absorption material at different milling time was obtained—see Figure 7. It can be seen that as the ball-milling time increased, and the $\mu^{\prime \prime }{\left(\mu^{\prime }\right)}^{-2}f^{-1}$ C0 value of the sample increased for the frequency range of 2–8 GHz. This indicates that the sample’s natural resonance-loss capacity at 4–6 GHz increased after ball milling. This was attributed to the enhanced anisotropy of the magnetic crystals after ball milling, which leads to the enhanced anisotropic field, the internal equivalent field enhancement of the ferromagnet, and the increased energy consumption generated by the damping effect. At 8–18 GHz, $\mu^{\prime \prime }{\left(\mu^{\prime }\right)}^{-2}f^{-1}$ remained basically unchanged, and the eddy-current loss was the main absorption mechanism at this time.

#### 3.4.2. Effects of Ball-Milling Time on Absorption

To study the absorption of materials, reflectivity was simulated using MATLAB software, based on the electromagnetic parameters measured by a vector network analyzer. The electromagnetic wave absorption capacity was expressed by the reflection loss RL (dB) as [20]
(8)$$Z_{in}=\sqrt{\frac{\mu_{r}}{\varepsilon_{r}}}\tanh\left[j\left(\frac{2\pi fd}{c}\right)\sqrt{\mu_{r}\varepsilon_{r}}\right]$$
(9)$$\mathrm{RL}\left(\mathrm{dB}\right)=20\log\left|\frac{Z_{in}-1}{Z_{in}+1}\right|$$


Here, h is the Planck constant, c is the speed of electromagnetic waves in vacuum, f is the frequency, d is the thickness of the material, Zin is the input impedance, $\mu_{0}$ and $\varepsilon_{0}$ are the permeability and permittivity of free space, $\varepsilon_{r}$ and $\mu_{r}$ are the permittivity and magnetic permeability of the material.

It can be seen from Figure 8 that the small spherical Fe_3_O_4_, without ball grinding, had a good absorption performance. When the matching thickness was 5.94 mm, it had the largest reflection loss capability at 3.84 GHz, and the reflection-loss reached −20.17 dB. When the matching thickness was 5.24 mm, it had the maximum reflection-loss capability at 15.28 GHz, reaching −41.25 dB. When the matching thickness was 5.98 mm, it showed the maximum reflection loss capability at 13.12 GHz, and the maximum reflection loss was 38.39 dB. This indicates that it had good absorption at both low and high frequencies.

However, as the ball-milling time increased, the absorption of Fe_3_O_4_ first decreased before it increased. This was because, when the hollow spherical Fe_3_O_4_ was ball milled, the hollow structure of the surface was destroyed, and the multiple reflection ability was reduced. Although the obtained Fe_3_O_4_ fragment particles were much finer, the overall electromagnetic wave attenuation performance still showed a significant decline. After ball milling for 0.5 h, the absorption performance of Fe_3_O_4_ decreased significantly, which suggest that the hollow spherical microstructure had a significant effect on the absorption of Fe_3_O_4_. With the extension of the ball-milling time, when the ball-milling times were 1 h and 1.5 h, a large number of Fe_3_O_4_ pellets were broken, the hollow structure severely damaged, and the dielectric loss continued to decrease. Thus, its maximum reflection loss at high frequencies was lower than −10 dB. However, as the ball-milling time continued to increase to 2 h, the spherical structure was almost ball-milled into finer nanoparticles. Because its nanoparticles were smaller in size and had a large specific surface area, it was beneficial to improve the absorption performance of Fe_3_O_4_. Therefore, compared with other ball grinding times, the absorption of Fe_3_O_4_ was significantly improved. When the matching thickness was 6.55 mm, there was a large reflection loss at 4.64 GHz, and the maximum reflection loss was −21.19 dB.

## 4. Conclusions

Effect of ball milling time on microstructure and absorption properties of Fe_3_O_4_ were investigated systematically. According to the above results, conclusions can be summarized at these following aspects.
The integrity and size of Fe_3_O_4_ particles all decreased as the ball milling time increased. The size of Fe_3_O_4_ particles decreased from 400 nm to 40 nm.The electromagnetic wave attenuation of hollow spherical Fe_3_O_4_ reduced by ball milling. More reflective interfaces and better conductive networks of hollow spherical Fe_3_O_4_ can be formed compared with smaller-sized nanoparticles. When the thickness of non-milled small spherical Fe_3_O_4_ absorption material was 5.24 mm, the reflection loss reached the maximum value −41.25 dB at 15.28 GHz.Fe_3_O_4_ showed excellent absorption properties as the milling time reached 2 h. When the thickness was 6.55 mm, the reflection loss reached the maximum of −21.19 dB at 4.64 GHz.


## Figures and Tables

**Figure 1 materials-13-00883-f001:**
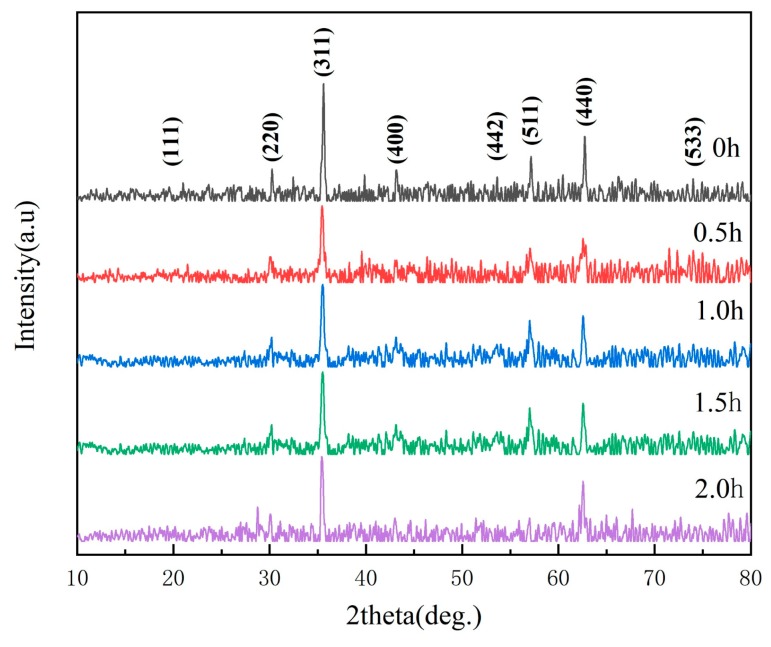
XRD spectra of Fe_3_O_4_ samples at different milling times.

**Figure 2 materials-13-00883-f002:**
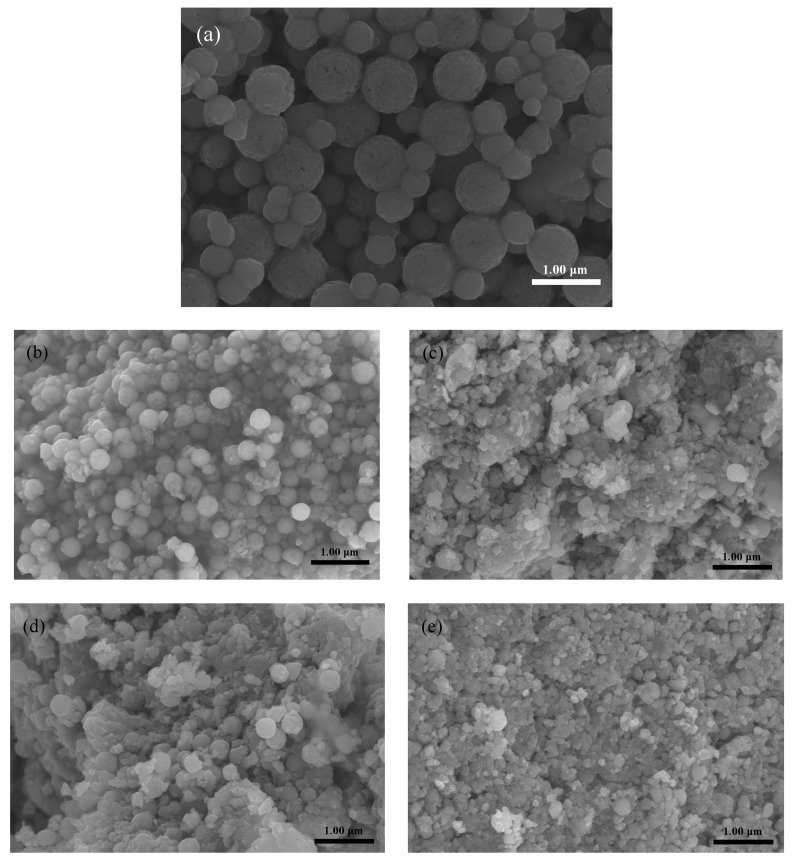
SEM images of Fe_3_O_4_ samples at different ball-milling times. (**a**) Ball milling for 0 h; (**b**) Ball milling for 0.5 h; (**c**) Ball milling for 1.0 h; (**d**) Ball milling for 1.5 h; (**e**) Ball milling for 2.0 h.

**Figure 3 materials-13-00883-f003:**
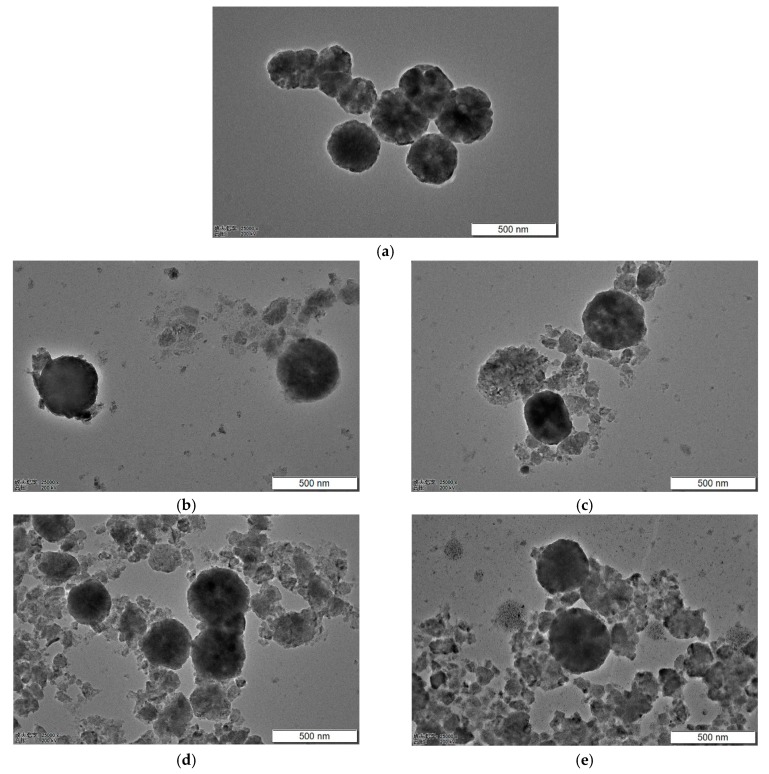
TEM images of Fe_3_O_4_ at different ball-milling times. (**a**) Ball milling for 0 h; (**b**) Ball milling for 0.5 h; (**c**) Ball milling for 1 h; (**d**) Ball milling for 1.5 h; (**e**) Ball milling for 2.0 h.

**Figure 4 materials-13-00883-f004:**
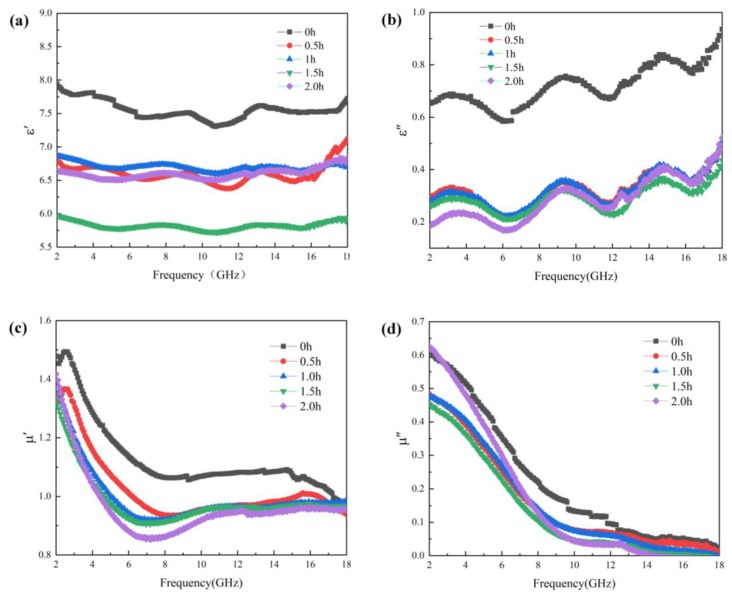
Complex permittivity and complex permeability of the Fe_3_O_4_ absorption material at different grinding times. (**a**) Real part of the complex permittivity; (**b**) Imaginary part of the complex permittivity; (**c**) Real part of the complex permeability; (**d**) Imaginary part of the complex permeability.

**Figure 5 materials-13-00883-f005:**
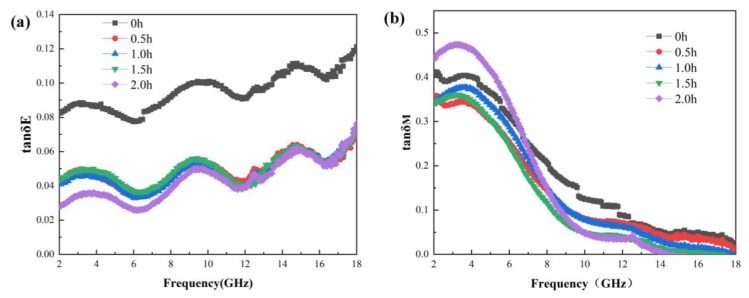
Electric loss tangent and magnetic loss tangent of the Fe_3_O_4_ absorption material at different grinding times. (**a**) Dielectric loss tangent; (**b**) Magnetic loss tangent.

**Figure 6 materials-13-00883-f006:**
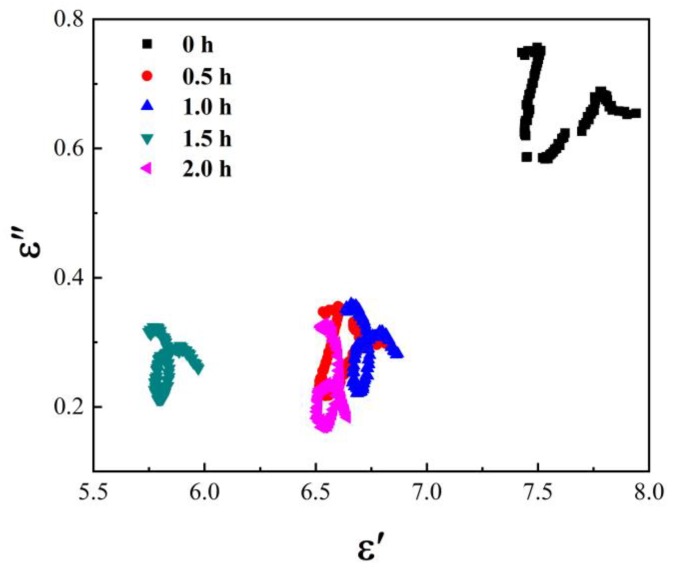
Cole–Cole diagram of the Fe_3_O_4_ absorption material at different ball-milling times.

**Figure 7 materials-13-00883-f007:**
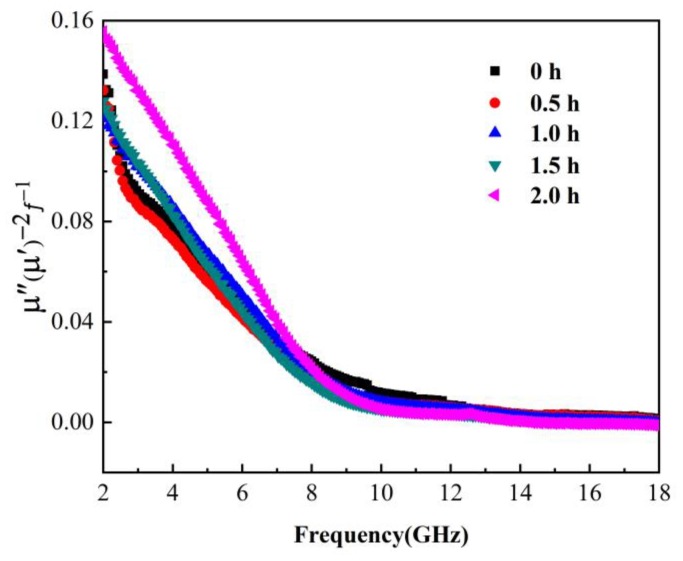
$\mu^{\prime \prime }{\left(\mu^{\prime }\right)}^{-2}f^{-1}$ vs. frequency curve of the Fe_3_O_4_ absorption material at different ball-milling times.

**Figure 8 materials-13-00883-f008:**
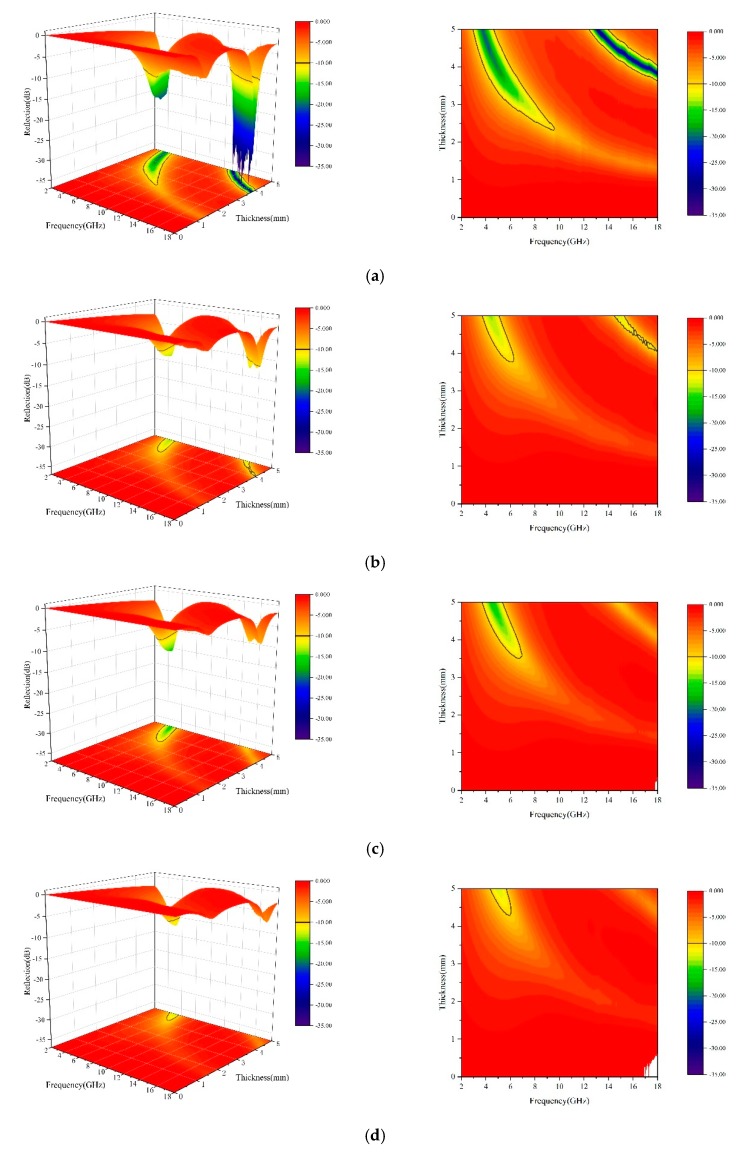
Three-dimensional reflectivity graphs of the Fe_3_O_4_ absorption materials at different ball milling times. (**a**) Ball milling for 0 h; (**b**) Ball milling for 0.5 h; (**c**) Ball milling for 1 h; (**d**) Ball milling for 1.5 h; (**e**) Ball milling for 2.0 h.

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
