# Peer review of "Effect of Ball Milling on the Absorption Properties of Fe3O4"

_materials, 2020, doi:10.3390/ma13040883_

Round 1

Reviewer 1 Report

I do not recommend publication of this manuscript in because it does not add any substantial physical content to the existing literature. That is not applied physics at the level we expect. This manuscript appears to contain a presentation of data without the in-depth analysis expected. Hence it is not sufficient to report outcomes of experiments without drawing out new physics in the interpretation of the results. In this manuscript, there are descriptions about the Figures, but there is little said about their physical significance and what new understanding can be gained from them. For example, what is the physics of FIg. 4? Unclear. There is no discussion of the graphs. For the real  part of the effective permittivity what is the physical origin of the wave nature of the curve (multiple peaks?): is that relaxation (of what?). Idem for the imaginary part of the permittivity (resonance? of what?). how the actual values compare with the bulk values? the varaition in Fig. 4(c) seems pretty weird to me (minimum at 7 GHz?). There are many modelling papers in the archival literature dealing with the  material parameter in the microwave range of frequencies: the authors should use them and fit to get physical conclusions. Fig. 6 is meaningless: not only the x-axis and y-axis should have same amplitude but no semi-circles are effective! Thus why using Cole-Cole odelling. Unclear. Overall, the authors are unable to explain the microwave absorption mechanisms in their material

Author Response

Dear Reviewer,

I sincerely thank your consideration and comments on the submitted paper “Effect of Ball Milling on the Absorption Properties of Fe3O4” (Manuscript ID: Materials-708173). Based on your comments and suggestions, we have made extensive modification on the original manuscript. Should you have any questions, please contact us without hesitate. Herein, I will reply the comment and suggestions raised by yours point by point, and the corresponding revisions with red marked are made in the paper.

I do not recommend publication of this manuscript in because it does not add any substantial physical content to the existing literature. That is not applied physics at the level we expect. This manuscript appears to contain a presentation of data without the in-depth analysis expected. Hence it is not sufficient to report outcomes of experiments without drawing out new physics in the interpretation of the results.

In this manuscript, there are descriptions about the Figures, but there is little said about their physical significance and what new understanding can be gained from them. For example, what is the physics of FIg. 4? Unclear. There is no discussion of the graphs. For the real part of the effective permittivity what is the physical origin of the wave nature of the curve (multiple peaks?): is that relaxation (of what?). Idem for the imaginary part of the permittivity (resonance? of what?). how the actual values compare with the bulk values? the varaition in Fig. 4(c) seems pretty weird to me (minimum at 7 GHz?). There are many modelling papers in the archival literature dealing with the material parameter in the microwave range of frequencies: the authors should use them and fit to get physical conclusions.

Thanks for your suggestion again, and your suggestion is quite good. As a traditional microwave absorbing material, Fe3O4 has high saturation magnetization and Curie temperature at high temperature (585 °C). This stable and excellent magnetic property has attracted many scholars’ attention. However, the high density of Fe3O4 limits its further use as a microwave absorbing materials (MAM). Fortunately, many studies have shown that hollowing of Fe3O4 is an effective way to reduce material weight, produce special morphological effects and improve electromagnetic properties. There are two ways to improve the present research level. One way is to prepare composite materials with ferrite and other materials to form wave absorbing materials with more complex microstructures; the other is to refine the structure size of materials to improve microwave absorption capacity. Therefore, in the present work, effect of ball milling time on microstructure, electromagnetic parameters, wave absorption properties and wave absorption mechanism of Fe3O4 by ball milling were investigated systematically. Especially the microwave attenuation mechanism and the absorption properties in different frequencies were analyzed and elaborated in detail. The results will be helpful for the researchers to understand the absorption mechanism, absorption properties and frequency characteristics of materials when preparing the absorbing material by ball milling. And it will be helpful to study absorption materials with lighter and higher properties.

At present, the measurement of electromagnetic parameters mainly uses a measurement method proposed by Nicolson and Ross [1] to measure the reflection transmission response of a transmission line filled with a substance to be measured. The magnetostatic properties were characterized by vibrating sample magnetometer (VSM, BHV-55). The permeability and permittivity of samples in the frequency range 2–18 GHz were tested by a vector network analyzer (VNA, N5242A, Agilent) for simulation of reflection loss. Composite sample were realized as follows: the wax was melted at 80 °C and mixed with the Fe3O4 powder homogeneously. The mixture was moved into a toroidal mold (Φin = 3.04 mm, Φout = 7 mm). The test software (Agilent, Santa Clara, CA, USA) is 85071 and the calibration part is 85050D. Before the test, the permittivity of air was measured as an evaluation of calibration effect.

Figure 1. Schematic representation of electromagnetic parameters measurement setup consisting of a vector network analyzer, a RF source, load match, and the toroidal shaped sample under test in the coaxial transmission waveguide. The inset picture in the dashed box shows the inner diameter, outer diameter and the height of as-obtained toroidal shaped sample [2]

[1]  Nicolson A.; Ross. G. Measurement of the intrinsic properties of materials by Time-Domain techniques. Transact. Instrument. Measurem. 1970, 19, 377-382.

[2]  Qiao M.; Lei X.; Ma Y.; Tian L.; Su K.; Zhang Q. Dependency of tunable microwave absorption performance on morphology-controlled hierarchical shells for core-shell Fe3O4@MnO2 composite microspheres. Chemi. Engin. J. 2016, 304, 552-562.

Figure 4 shows the curves of the complex permittivity and complex permeability of the Fe3O4. Figure 4(a) and (b) show the real part imaginary part of the complex permittivity of the absorption material. The real part of the complex dielectric constant is generated by various displacement polarizations in the material, and represents the charge energy storage capacity of the material. The imaginary part of the complex dielectric constant is generated by various relaxations caused by the turning polarization, and represents the charge energy loss capacity of the material. The different dielectric constants of ferric oxide with different ball milling treatments are quite different. The real and imaginary parts of the complex permittivity of the sample are larger as the ball milling time is 0 h. Due to the increased contact area, a large number of hollow Fe3O4 spheres formed a macroscopic conductive chain or local conductive network in the material, under the action of the electromagnetic field. Therefore, in the absence of ball grinding, the absorption material with a large amount of Fe3O4 hollow pellets, had both a higher electrical conductivity and dielectric constant. The structure of Fe3O4 was destroyed and the hollow advantage was lost, the real and imaginary parts decreased as the milling time increased. When the ball milling time was long enough, Fe3O4 gradually became a nanostructure. Due to the special quantum effect, Fe3O4 has a large specific surface area, surface polarization and dipole polarization increase, so the real and imaginary parts of the dielectric constant become larger. Figure 4(c) and (d) show the real and imaginary parts of the complex permeability of the Fe3O4 absorption. Compared with the large-scale change of the complex dielectric constant, the morphologies of Fe3O4 have a much smaller effect on the permeability characteristics. It can be observed that the real part of the complex magnetic permeability of the Fe3O4 shows a similar change, and the initial real part values are all greater than 1, which is attributed to the ferromagnetism of the Fe3O4. As the frequency of the electromagnetic field increases, the real part value keeps decreasing, and then drops below 1, which can be attributed to the diamagnetism of ferric oxide at high frequencies caused by eddy current loss.

Fig. 6 is meaningless: not only the x-axis and y-axis should have same amplitude but no semi-circles are effective! Thus why using Cole-Cole odelling. Unclear. Overall, the authors are unable to explain the microwave absorption mechanisms in their material.

Your suggestion is good. Figure 6 shows the relationship between the imaginary part and the real part of the dielectric constant of Fe3O4 by the Cole-Cole model. The relationship between ε′ and ε″ will be presented in a semicircle as in Cole-Cole diagram. If the dielectric constant polarization is generated by the relaxation process, it will be presented in the form of a Debye semicircle. The Debye semicircle can reflect the polarization spectral characteristics of the medium and the degree of relaxation. Based on the above theories, Cole-Cole diagrams (2D) of the Fe3O4 absorption material, at different ball milling times, can be obtained from Figure 6. Firstly, there are multiple Debye semicircles in all samples, indicating that there are multiple polarization relaxation processes at different frequencies. According to formula 6, the optical frequency permittivity ε, static permittivity εs and the conductivity σ = σr + σR can be calculated by the center and radius of the semicircle. Therefore, the Debye semicircle radius and abscissa values of Fe3O4 are relatively large, leading to the higher ε, εs and σ.

We would like to express our great appreciation to you. Looking forward to hearing from you.

Thank you and best regards.

Yours sincerely,

Yujiang Wang

Shicheng Wei

Bo Wang

Corresponding authors

Address:     Dr. Yujiang Wang

                 Pro. Shicheng Wei

                 Dr. Bo Wang

National Key Laboratory for Remanufacturing,

Academy of Army Armored Forces, Beijing 100072,

People’s Republic of China

            Tel: +86-010-66718541; +86-010-66719083; +86-010-66718540

            E-mail: yjwang201617@163.com; scwei55555@163.com;

                        wangbobo421@163.com

Reviewer 2 Report

The manuscript is well written, the magnetic materials and their application are of interest. I think it could be taken into consideration for publication in Materials.

 Please review the manuscript for typos. i.e. L135, L145

Please rewrite the conclusions.

Author Response

Dear Reviewer,

I sincerely thank your consideration and comments on the submitted paper “Effect of Ball Milling on the Absorption Properties of Fe3O4” (Manuscript ID: Materials-708173). Based on your comments and suggestions, we have made extensive modification on the original manuscript. Should you have any questions, please contact us without hesitate. Herein, I will reply the comment and suggestions raised by yours point by point, and the corresponding revisions with red marked are made in the paper.

The manuscript is well written, the magnetic materials and their application are of interest. I think it could be taken into consideration for publication in Materials.

Please review the manuscript for typos. i.e. L135, L145

Please rewrite the conclusions

Thanks for your suggestion. I am sorry for my careless. According to your suggestion, relevant types have been revised, and the conclusions have been rewritten.

We would like to express our great appreciation to you. Looking forward to hearing from you.

Thank you and best regards.

Yours sincerely,

Yujiang Wang

Shicheng Wei

Bo Wang

Corresponding authors

Address:     Dr. Yujiang Wang

                 Pro. Shicheng Wei

                 Dr. Bo Wang

National Key Laboratory for Remanufacturing,

Academy of Army Armored Forces, Beijing 100072,

People’s Republic of China

            Tel: +86-010-66718541; +86-010-66719083; +86-010-66718540

            E-mail: yjwang201617@163.com; scwei55555@163.com;

                        wangbobo421@163.com

Reviewer 3 Report

The manuscript describes the effect of milling on the absorption property in Fe3O4. The study is highly applicative since the fabrication of Fe3O4 powder of good electromagnetic properties is hot industrial topic. The manuscript is easy to read even for the non specialist and all experimental protocols are well described. 

I have no scientific concern regarding the manuscript which should be accepted for publication.

However the quality of the figures should be improuve. The resolution is pour and the axes are hard to read.

Author Response

Dear Reviewer,

I sincerely thank your consideration and comments on the submitted paper “Effect of Ball Milling on the Absorption Properties of Fe3O4” (Manuscript ID: Materials-708173). Based on your comments and suggestions, we have made extensive modification on the original manuscript. Should you have any questions, please contact us without hesitate. Herein, I will reply the comment and suggestions raised by yours point by point, and the corresponding revisions with red marked are made in the paper.

The manuscript describes the effect of milling on the absorption property in Fe3O4. The study is highly applicative since the fabrication of Fe3O4 powder of good electromagnetic properties is hot industrial topic. The manuscript is easy to read even for the non specialist and all experimental protocols are well described.

I have no scientific concern regarding the manuscript which should be accepted for publication.

However the quality of the figures should be improved. The resolution is poor and the axes are hard to read.

Thanks for your suggestions. According to your suggestion, the quality of the figures has been improved.

We would like to express our great appreciation to you. Looking forward to hearing from you.

Thank you and best regards.

Yours sincerely,

Yujiang Wang

Shicheng Wei

Bo Wang

Corresponding authors

Address:     Dr. Yujiang Wang

                  Pro. Shicheng Wei

                  Dr. Bo Wang

National Key Laboratory for Remanufacturing,

Academy of Army Armored Forces, Beijing 100072,

People’s Republic of China

            Tel: +86-010-66718541; +86-010-66719083; +86-010-66718540

            E-mail: yjwang201617@163.com; scwei55555@163.com;

                       wangbobo421@163.com

Round 2

Reviewer 1 Report

I stand by my previous opinion. The part on the electromagnetic measurements is not interpreted corrected and is completely fuzzy. It does not bring any innovation in terms of materials and physics. Fig. 6 is completely useless (where are the semi-circles here?). Overall, the authors are unable to gve the reader any physical explanation of the absorption mechanism in the microwave range of frequencies.